# Intestinal Ischemia Secondary to Blunt Abdominal Trauma in Late Pregnancy: A Case Report of a Rare Complication with Serious Implications

**DOI:** 10.3390/jcm14165808

**Published:** 2025-08-16

**Authors:** Marta Domínguez-Moreno, Ana María Ferrete-Araujo, Mónica Marín-Cid, Juan José Egea-Guerrero, Lucas Cerrillos

**Affiliations:** 1Department of Materno-Fetal Medicine, Genetics and Reproduction, Virgen del Rocío University Hospital, 41013 Seville, Spain; cidmmonica@gmail.com (M.M.-C.); lucascerrillosg@gmail.com (L.C.); 2Intensive Care Unit, Rehabilitation and Traumatology Hospital, Virgen del Rocío University Hospital, 41013 Seville, Spain; amferretearaujo@gmail.com (A.M.F.-A.); jjegeaguerrero@gmail.com (J.J.E.-G.); 3Department of Medicine, University of Seville, 41004 Seville, Spain

**Keywords:** blunt abdominal trauma, pregnant woman, intestinal ischemia, medical emergency

## Abstract

**Background**: Blunt abdominal trauma in pregnancy is a medical emergency with significant maternal-fetal morbidity and mortality. Although rare, intestinal ischemia can occur as a serious abdominal complication following trauma during pregnancy. **Case presentation**: A 41-year-old woman at 33 weeks and 6 days of gestation was involved in a car accident, as a passenger in the front seat of a vehicle that left the road and overturned. The initial examination revealed severe chest trauma but no immediate signs of abdominal injury. However, the patient’s condition worsened, showing delayed symptoms of gastrointestinal dysfunction, clinical deterioration, and labor onset. Complementary imaging studies did not reveal conclusive findings suggesting complications related to the blunt abdominal trauma. Following a multidisciplinary team’s decision to perform an emergency cesarean section in the maternal-fetal interest, intestinal ischemia secondary to a mesenteric tear was discovered, necessitating intestinal resection and end-to-end anastomosis. **Conclusions**: Despite being a rare condition often associated with diagnostic delays, in cases of sudden clinical deterioration or maternal hemodynamic instability, immediate multidisciplinary intervention is essential. This approach may allow the early detection of trauma-related complications, reducing potentially preventable deaths and achieving favorable maternal and neonatal outcomes.

## 1. Introduction

Traumas involving pregnant women are uncommon but potentially serious, as both the mother and fetus face significant risks [1]. Blunt abdominal trauma is the leading non-obstetric cause of maternal mortality [2], with road traffic accidents (55%), falls (22%), and assaults (22%) being the most common causes [3,4,5,6,7]. The precise incidence is difficult to determine, as most available data come from observational and retrospective studies or case series [8]. It is estimated that one in twelve pregnancies could be complicated by trauma [8], especially in the third trimester [7].

Blunt abdominal trauma is considered a medical emergency due to its association with significant maternal and fetal morbidity and mortality [7,9,10,11]. Reported mortality rates vary widely, from 13% to 51% [5,12], and pregnant women face almost double the mortality risk compared to non-pregnant trauma patients [13]. In fact, 50% of maternal deaths during pregnancy are attributed to trauma [14].

Gestational trauma has substantial implications for the fetus as well [8]. Although life-threatening trauma accounts for less than 8% of all traumas, it carries a 40% to 50% risk of fetal loss [1,15] due to complications such as premature rupture of membranes, premature labor, uterine rupture, placental abruption, direct fetal injury, or maternal shock [8,15]. Fetal loss is more likely to occur as the severity of the trauma increases. For example, it is estimated that at least 3.7 fetal and neonatal deaths per 100,000 pregnancies are related to automobile accidents [16], with placental abruption being the second most common cause of post-traumatic fetal death when the mother survives [8].

Thus, the early detection and management of trauma-related complications require multidisciplinary care at a trauma referral center. This team should include specialists from emergency medicine, obstetrics, gynecology, general surgery, and intensive care.

The aim of this article is to present a case of blunt abdominal trauma in a pregnant woman in her third trimester complicated by intestinal ischemia. After a thorough literature review, we found no similar cases previously reported.

## 2. Case Presentation

A healthy 41-year-old pregnant patient at 33 weeks and 6 days of gestation, with no relevant medical or surgical history, was admitted after a traffic accident. She was seated in the front passenger seat of a four-wheeled vehicle, wearing an adapted seat belt, when the vehicle left the road and overturned. After initial on-site assistance by emergency services, she was found to have respiratory distress with severe pain in the right hemithorax and moderate hypoxemia. She was hemodynamically stable. Initial vitals were blood pressure 100/60 mmHg, heart rate 95 beats per minute, 87% oxygen saturation. No hypotension or neurological deterioration was observed (Glasgow Coma Scale score 15 out of 15).

The patient was transferred to a level-one trauma referral center equipped for obstetric care [7,16]. Physical examination revealed subcutaneous emphysema in the right hemithorax and paradoxical breathing. A chest X-ray showed a right pneumothorax, prompting the placement of emergent endothoracic drainage due to multiple fractures of the right anterior costal arches (from the second to the seventh ribs). This accounted for both the right pneumothorax and the flail chest. The pelvis was stable, and there was no abdominal pain on palpation or genital bleeding. Superficial injuries, including abrasions, a small supraumbilical hematoma, and another hematoma on the lateral wall of the upper abdomen, were noted, with no clear signs of seat belt injury.

In the initial Focused Assessment with Sonography for Trauma (FAST), no solid-organ injuries or free fluid in the Morrison space, splenorenal space, or Douglas pouch were observed. A gynecological assessment via obstetric ultrasound confirmed fetal vitality and normal placental positioning, with no evidence of placental abruption. Fetal monitoring showed no pathological patterns, and there was no uterine activity on admission.

During the first three days in the Intensive Care Unit (ICU), the patient’s condition remained stable. She received optimal pain control with intravenous analgesia, and her vital signs, as well as blood analyses, remained consistent throughout her stay. The evolution of her vital parameters during hospitalization is shown in Table 1.

After this period, the patient began to experience uterine activity, which was initially treated with tocolytics. Pulmonary maturation was also induced due to her gestational age, in anticipation of potential preterm delivery.

On the fourth day of admission, there was an abrupt deterioration. The patient developed severe gastrointestinal symptoms, including intense epigastric pain, heartburn, nausea, and recurrent vomiting, which complicated food intake. These symptoms persisted despite antacid treatment. On examination, her abdomen was distended, firm, and diffusely tender to palpation. Constipation was noted, despite administering a cleansing enema.

Given the suspicion of paralytic ileus, a conservative approach was initially taken, with nasogastric decompression resulting in partial improvement and evacuation of a significant volume of fecal content. Non-invasive imaging studies were requested, including an obstetric ultrasound to verify fetal viability and an abdominal ultrasound to investigate potential intra-abdominal trauma-related complications. The obstetric ultrasound revealed a fetus in cephalic presentation with normal heart rate and movements. The abdominal ultrasound showed a significantly distended stomach, but no other pathological findings.

Laboratory results showed no significant changes since admission. The evolution of the principal findings is detailed in Figure 1.

Given the marked clinical worsening of the patient, an arterial blood gas analysis was performed, showing pH 7.3, arterial CO_2_ pressure of 29 mmHg, and HCO_3_ of 18 mEq/L, indicating metabolic acidosis, likely due to repeated vomiting and tachypnea.

As the patient’s hemodynamic and respiratory condition deteriorated, clinical signs such as tachycardia, tachypnea, and moderate hypoxemia were noted (detailed in Table 1). Combined with the suspicion of intestinal pseudo-obstruction, emergency medical specialists, in agreement with obstetrics specialists, decided to pursue a more accurate and relevant diagnostic imaging test. After weighing the risk–benefit ratio with the Radiology Department and minimizing fetal risk, a thoraco-abdomino-pelvic computed tomography (CT) was performed. The CT scan revealed no solid organ injury but showed significant gastric distension and dilated intestinal loops at the level of the duodenum, Treitz ligament, and jejunum. There were no signs of loop ischemia, intestinal obstruction, perforation, or mesenteric vessel thrombosis (Figure 2).

Following the CT, when the patient was transferred back to the ICU, she began experiencing regular and painful uterine contractions. Given the clinical instability and high suspicion of acute abdomen, and after verifying fetal viability in a near-term pregnancy, a multidisciplinary team (including intensivists, obstetricians, and general surgeons) reached a consensus to perform an emergency cesarean section. Vaginal delivery was ruled out due to concerns for maternal-fetal well-being and the potential for unresolved obstetric complications.

A Pfannenstiel incision was made, following standard procedure. The fetal extraction was uneventful, resulting in the birth of a newborn weighing 1985 gr, showing good vitality, and scored 9–10 on Apgar Test. The umbilical artery pH was 7.07. He only required aspiration of slightly fecal-like secretions. No additional care beyond the usual was needed.

After the birth of a live newborn, the abdominal cavity was explored, and the incision was extended infraumbilically. This revealed a serous-purulent collection, consistent with secondary peritonitis affecting all four abdominal quadrants. Additionally, a small bowel loop at the ileal level, approximately 10 cm from the ileocecal valve, showed signs of ischemic damage, with a 15 cm necrotic section identified. An intestinal resection was performed on the affected loop, followed by end-to-end anastomosis.

In the postoperative period, the patient’s clinical course was favorable. She was discharged home two weeks after surgery without any postoperative complications. The child was re-evaluated at the one-month postnatal follow-up. Good weight gain and achievement of milestones appropriate for baby’s age were confirmed.

The patient’s complete clinical evolution from admission to discharge is detailed in the Figure 3.

## 3. Discussion

The management of severely polytraumatized patients requires well-established protocols to ensure a rapid, coordinated response that leads to favorable functional and survival outcomes. When the patient is a pregnant woman, as in the case described, the complexity of management increases significantly, necessitating a multidisciplinary approach involving various medical and surgical specialties to ensure both maternal and fetal safety.

The anatomical and physiological changes during pregnancy complicate the assessment of traumatic injuries [14,15,17], particularly after 12 weeks of gestation [18]. These changes may also affect responses to resuscitation and diagnostic test results. As gestation progresses, the growing uterus and fetus increase the risk of fetal and placental injury, with the third trimester being the period of highest risk [19].

The primary obstetric concern in cases of blunt trauma is the tension and displacement of the uterus during impact, which can lead to placental abruption due to backblow mechanisms [8]. Direct fetal injury may occur due to blunt force impact, such as hitting the steering wheel in a car accident or due to intracranial trauma associated with pelvic fractures when the fetus is in a cephalic presentation during the third trimester. Additionally, the growing uterus pushes the intestinal loops upwards, making the intestines one of the more protected organs from blunt trauma, though this was not the case in our patient. The presence of abrasions in the right upper abdominal quadrant during the initial physical exam could have indicated possible blunt abdominal trauma.

Unlike penetrating trauma, blunt trauma lacks obvious external signs, making it difficult to gauge the severity of injuries, which can result in delayed diagnoses of severe, life-threatening intra-abdominal injuries [4]. Moreover, clinical signs of peritoneal irritation are less obvious in pregnant women, reducing the effectiveness of physical exams in detecting such injuries.

In pregnancy, the hyperdynamic and hypervolemic state can mask significant blood loss [2,14]. A pregnant woman may not exhibit typical signs of shock even after losing up to 20% of her blood volume [7]. Only after blood loss exceeds 2500 milliliters may the mother’s condition rapidly decline, reducing the chances of fetal survival to 20% [3].

Healthcare providers face the challenge of managing two patients simultaneously: the mother and the fetus [17]. A multidisciplinary approach is critical to optimizing care, and the main focus must remain on stabilizing the mother since fetal outcomes are directly linked to maternal survival and stability [8]. What is beneficial for the mother is beneficial for the fetus [8,15].

After maternal stabilization, radiological tests (e.g., chest and pelvic X-rays, computed tomography) should be performed if necessary and should not be delayed due to unfounded concerns about fetal effects [2,8]. Studies show that these tests can identify significant, silent lesions in symptomatic patients, even when abdominal ultrasound results are inconclusive [8,18,20]. When ionizing radiation is required, the uterus should be shielded, and it is important to note that radiation doses below 5 rads (0.05 Gy) are not associated with increased risks of anomalies, spontaneous abortion, or fetal growth restriction [14,15]. CT scanning remains the fastest and most accessible diagnostic tool for abdominal and pelvic trauma [21].

Treatment for blunt abdominal trauma in pregnant patients who are hemodynamically stable is typically conservative [14,22]. However, early surgical intervention may be necessary in cases of hemodynamic instability or when bowel injury is highly suspected. Hypotension and sepsis are particularly dangerous and can be fatal to both the mother and the fetus [14].

Gastrointestinal injuries in the context of blunt trauma are rare, with a prevalence of only 1% among trauma admissions in the general population [23]. The most commonly affected hollow organ is the small intestine, and injuries in this location can be silent, delaying diagnosis and treatment and significantly increasing maternal and fetal morbidity and mortality [23]. The initial diagnosis of hollow viscera injury relies on physical exam findings and CT imaging, with sensitivities between 55% and 95%, and specificities ranging from 48% to 92%, and a false negative rate of up to 15% [23]. In cases of diagnostic uncertainty, urgent surgery should be considered.

In specific situations where maternal deterioration is imminent, as in this case, emergency cesarean sections should be considered to safeguard both maternal and fetal life. Studies report that, in pregnant women beyond 25 weeks of gestation [3], cesarean section performed for trauma-related indications results in a 45% fetal survival rate and a 72% maternal survival rate [3]. This knowledge is crucial to making the correct multidisciplinary decision, which can lead to optimal maternal and neonatal outcomes, as demonstrated in this case.

## 4. Conclusions

Intestinal ischemia following blunt abdominal trauma in a pregnant patient is a rare and difficult-to-diagnose condition. Delayed treatment can result in significant morbidity and mortality for both mother and fetus. Rapid surgical intervention through cesarean section and exploratory laparotomy is necessary in cases of clinical deterioration and maternal instability to identify and treat life-threatening injuries. Coordinated multidisciplinary care is key to early detection of silent complications, potentially preventing avoidable deaths and achieving favorable maternal and neonatal outcomes.

## Figures and Tables

**Figure 1 jcm-14-05808-f001:**
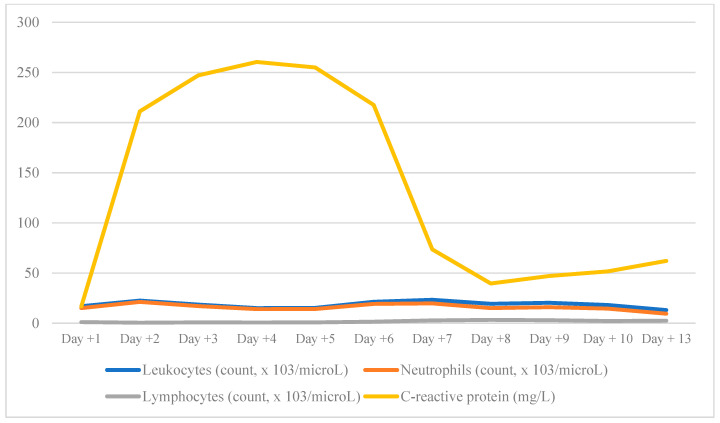
Evolution of principal laboratory findings during hospital stay.

**Figure 2 jcm-14-05808-f002:**
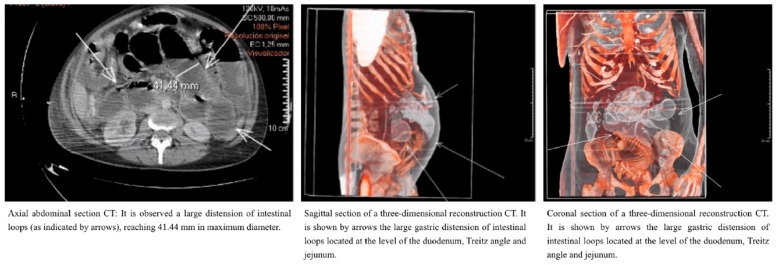
Axial, sagittal and coronal section CT.

**Figure 3 jcm-14-05808-f003:**
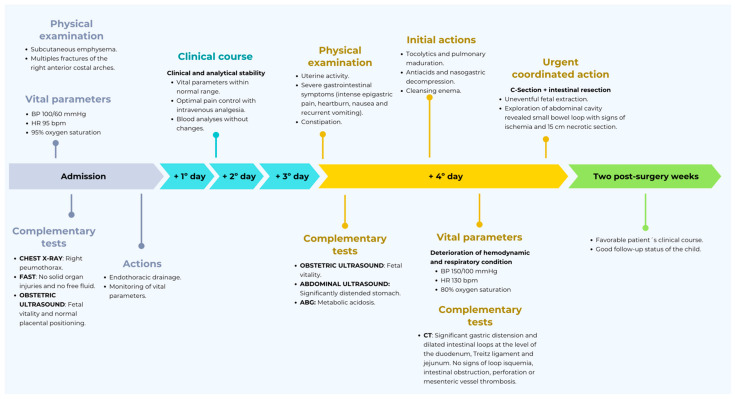
Patient’s complete clinical evolution during hospital stay.

**Table 1 jcm-14-05808-t001:** Evolution of vital parameters during hospital stay.

Days of Admission	Day +1	Day +2	Day +3	Day +4	Day +5	Day +6	Day +7	Day +8	Day +9	Day +10	Day +11	Day +12	Day +13
Blood pressure (mmHg)	110/60	100/60	110/60	150/100	110/55	110/60	110/55	110/65	110/60	112/65	109/55	110/64	110/55
Heart rate (beats per minute)	95	100	110	130	95	70	74	81	75	78	77	80	76
24 h Diuresis (mL)	2400	2590	1570	1400	1300	2160	2000	2050	2170	2140	2200	2130	2100
Oxygen saturation (%)	95	95	95	80	98	99	99	98	98	99	99	99	99
Pleural Drainage (mL)	Tube insertion	400 (slightly haematitic)	800 (slightly haematitic)	50	Clamped	Removed							
Abdominal Drainage (mL)					30 (abdominal, sero-hematitic)	20 (serous)	20 (serous)	15 (sero-hematitic)	Removed				

## Data Availability

The datasets used and analyzed during the current study are available from the corresponding author on reasonable request.

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
