# Peer review of "Intestinal Ischemia Secondary to Blunt Abdominal Trauma in Late Pregnancy: A Case Report of a Rare Complication with Serious Implications"

_jcm, 2025, doi:10.3390/jcm14165808_

Round 1
Reviewer 1 Report
Comments and Suggestions for Authors
Reviewer
Initial comments
This Case report demonstrates that physical examination and clinical observation are essential for an accurate diagnosis and appropriate treatment to save the lives of both the pregnant woman and the fetus.
This case was very well managed, with a satisfactory outcome. When the cesarean section was performed, we understood that the fetus was alive, but the condition of the fetus when it was removed was not specified.
“ After the birth of a live newborn, the abdominal cavity was explored, and the incision was extended infraumbilically”.
In this case, the child's APGAR score was missing, as the mother's condition is very well documented.
- What is the main question addressed by the research?
The main issue addressed in this case report is the importance of rapid and accurate diagnosis, as the complication observed in this patient at the end of pregnancy, in addition to the thoracic trauma that was being treated, was the diagnosis of “Intestinal ischemia secondary to blunt abdominal trauma in a pregnant woman” which was very serious and if not treated as reported by the authors, the prognosis would be very poor and the outcome of this case could have been fatal.
-Do you consider the topic original or relevant to the field? Does it address a specific gap in the field? Please also explain why this is/ is not the case.
Yes, very important, as the indication of surgery by a multidisciplinary team enhances the diagnosis and encourages the indication for pregnant patients even at the end of pregnancy. This case report does not address a “specific gap in the field,” even because accidents of this magnitude are never expected in pregnant patients, although they can happen, and health professionals must be prepared for these types of emergencies, not just obstetricians. This is why the multidisciplinary team is so important, as it is a very important aspect of this case report.
-What does it add to the subject area compared with other published material?
There are few case reports in the field of obstetrics that mention accidents that caused these complications, which fortunately, with appropriate management, were managed in the best possible way. An extremely relevant fact in this case was the restored health of both the mother and the child, who was born healthy (as I mentioned in the opening remarks, reports of the child's condition, such as the APGAR score, were missing).
-What specific improvements should the authors consider regarding the methodology?
I believe the methodology was satisfactory in this case report.
-Are the conclusions consistent with the evidence and arguments presented and do they address the main question posed? Please also explain why this is/is not the case.
Yes, the conclusions are in line with the objectives of this case report.
~ Are the references appropriate?
Yes, there are 23 references relevant to the paper.
~ Any additional comments on the tables and figures.
Tables Table 1. Evolution of vital parameters during hospital stay. and 2 Table 2. Evolution of laboratory findings during hospital stay. are very well placed as well as Figure 1. Figure 1. Axial, sagital and coronal section TC.
Thank you
Author Response
Thank you for all your comments. Following it, we have modified the original manuscript.
1.When the cesarean section was performed, we understood that the fetus was alive, but the condition of the fetus when it was removed was not specified. The child´s APGAR score was missing, as the mother´s condition is very well documented.
1.As suggested, we have added information related to child´s condition after fetal extraction. Following the standard protocol implemented in our center, in case of ending the pregnancy by cesarean section, a well-structured neonatology team comes to the operating theatre to attend the newborn and to initiate the basic neonatal care. Fetal vitality is checked and wellbeing tests carrying out by Apgar test at 1 and 5 minutes. In addition, the newborn is weighed and measured, and a sample of umbilical cord blood is taken to determine the umbilical artery pH at birth. All these actions were implemented in the present clinical case.
Specifically, the newborn weighed 1985 gr (normal weight), showed good vitality at birth, and scored 9-10 on Apgar test. The umbilical artery pH was 7.07. He only required aspirations of slightly fecal-like secretions. No additional care beyond the usual was required, nor was necessary the admission to the neonatal care unit. In the postnatal follow-up, one month after birth, good weight gain and achievement of milestones appropriate for the baby´s age were confirmed. All this information has been included in the Main Text.
Reviewer 2 Report
Comments and Suggestions for Authors
Dear Author,
I have read the manuscript "Intestinal ischemia secondary to blunt abdominal trauma in a pregnant woman: A rare complication with serious implications." with great interest.
Title
The title should report information regarding the nature of the manuscript (e.g., " A case report", "A systematic review")
Introduction
You reported a lot of epidemiological data. What about the relationship between abdominal trauma and bowel ischemia in non-pregnant patients?
What about the relationship between abdominal trauma and bowel ischemia in pregnant patients? And related to this second question, what does your manuscript add to the existing literature?
Case presentation
What about the patient's medical history? What about BMI?
What about vital parameters on admission? What about blood tests on admission (e.g., ABG)? What about GCS on admission?
What about lactates on admission and in the following days?
What about the outcome for the fetus?
Discussion
What about MRI?
Do you have any considerations regarding the strengths and limitations of your approach to this case?
What about early(er)-recognition of the bowel complication? Was it possible?
Author Response
Thank you for all your comments. Following it, we have modified the original version of the manuscript.
1.The title should report information regarding the nature of the manuscript (e.g. “A cases report”, “A systematic review”).
1.Thank you to the reviewer for the suggestion to describe precisely in the title the nature of the manuscript. Following it, we have edited the original title to this one: “Intestinal ischemia secondary to blunt abdominal trauma in late pregnancy: A case report of a rare complication with serious implications”.
2.You reported a lot of epidemiological data. What about the relationship between abdominal trauma and bowel ischemia in non-pregnant patients?
What about the relationship between abdominal trauma and bowel ischemia in pregnant patients? And related to this second question, what does your manuscript add to the existing literature?
2.We did not wish to delve into the relationship between abdominal trauma and intestinal ischemia in the general population, as this was not the focus of our clinical case.
Regarding the connection between abdominal trauma and intestinal ischemia in the specific group of pregnant women, it is discussed in detail the “Discussion” section. As you can read, the physiological and anatomical changes that can occur during pregnancy and may hinder the initial approach to the pregnant patient are detailed. Likewise, the possible mechanisms of injury and the clinical signs and symptoms that may increase our diagnostic suspicion are explored in depth. The complementary tests to be performed and the different options of treatment in each case are also presented.
We consider that this manuscript highlights the importance of keeping in mind unusal diagnosis in pregnant women who are transferred to the emergency department after abdominal trauma, as well as the crucial role of a close monitoring and multidisciplinary follow-up among members of different specialties for optimal case management, with the aim of achieving the best possible obstetric and perinatal outcomes.
3.What about the patient´s medical history? What about BMI?
3.It is true that in the manuscript we make no mention of the patient's medical history. The reason behind this, it is that the patient had no medical or surgical history of note. She was a healthy lifestyle patient, aged 41 years, BMI of 24 kg/m2 and with no toxic habits. She had been under good gestational control until admission. Nevertheless, in order to clarify the patient's medical condition prior to admission, we have included a brief description in the “Case Presentation” section.
4.What about vital parameters on admission?
4.As detailed in the “Case Presentation” section, no hypotension or neurological deterioration was observed. In particular, on admission, the patient was hemodynamically stable (BP 100/60 mmHg, HR 95 bpm, 87% oxygen saturation) and afebrile. She was conscious, oriented in time and space, and cooperative during the interview.
5.What about blood tests (e.g. ABG -arterial blood gas) on admission?
5.Unfortunately, the ABG on admission is not included in the digital medical record. It is recorded on a paper form and kept in an envelope with all the patient's documentation in the hospital's Documentation Unit. If the reviewer considers this information to be crucial, we could request it from the aforementioned Unit. However, we would like to point out that the process may take a considerable amount of time and could possibly delay publication.
6.What about GCS on admission?
6.As stated in the text, the patient showed no neurological deterioration. It was reflected in a Glasgow Coma Scale score of 15 out of 15.
7.What about lactates on admission and in the following dates?
7.Unfortunately, the lactate values during the patient's admission are not included in the digital medical record. They are recorded on a paper form and kept in an envelope with all the patient's documentation in the hospital's Documentation Unit. If the reviewer considers this information to be crucial, we could request it from the aforementioned Unit. However, we would like to point out that the process may take a considerable amount of time and could possibly delay publication.
8.What about the outcomes for the fetus?
8.Following your recommendations and those of editorial consultant 1, we have expanded the information provided on the condition of the newborn at birth and his follow-up status at discharge.
9.What about MRI?
9.Unfortunately, MRI is not available at our center for urgent testing and may take days to perform it. In situations of extreme urgency such as the one described; the additional imaging test available is CT. It provides a good definition of abdominal viscera and can help to make clinical decisions as in this clinical case, without posing a high risk to the fetus.
10.Do you have any considerations regarding the strengths and limitations of your approach to this case?
10.We could add as a strength the fact that we have a multidisciplinary team that works in a coordinated manner to solve complex cases. Another great advantage of the case was that it took place in a tertiary level hospital that has the necessary material resources to take medical actions without delay.
As limitations, we could consider the delay in diagnosis, not because of the lack of medical suspicion of knowledge, but due to the pregnancy itself. It can make difficult to identify warning signs until advanced stages of the medical or surgical complication.
11.What about early(er)-recognition of the bowel complication? Was it possible?
As described in the “Discussion” section the anatomical and physiological changes during pregnancy complicate the assessment of traumatic injuries.
Given the initial clinical stability of the patient expressed as vital parameters in normal range, the optimal pain control and the low risk of hidden injuries given the accident report (vehicle overturning at low speed, no data on complications, no loss of consciousness or difficulty in extracting the patient), together with the absence of warning signs in the complementary tests performed on admission (simple X-ray and abdominal ultrasound), no additional imaging tests such as CT scan were not indicated. For all these reasons, a conservative management was initially chosen.
Unfortunately, the patient's hemodynamic status and the serious clinical condition without improvement despite initial conservative actions (a significant abdominal distension, the lack of clinical improvement despite gastric decompression and abundant fecal contents), made necessary to take an urgent interventional decision, given the suspect of a serious complication.
This decision was decision was made without delay, leading to optimal results for both the mother and the newborn. That is why we agree that the urgency of the situation was quickly recognized ant the right decisions were made.
Reviewer 3 Report
Comments and Suggestions for Authors
Manuscript entitled “Intestinal Ischemia Secondary to Blunt Abdominal Trauma in a Pregnant Woman: A Rare Complication with Serious Implications” by Domínguez‑Moreno M. et al. (case report)
This is a well‑written and clinically instructive report of a third‑trimester gravida who, after a motor‑vehicle collision, developed delayed small‑bowel ischemia from a mesenteric tear that was only discovered at emergent caesarean section.
Comments:
- The manuscript’s narrative regarding the clinical course lacks temporal clarity, alternating between references such as “day +4” and “after 72 hours” without a coherent sequence of key events. The authors should provide a clear timeline figure or table that maps hours post-injury against clinical events, including the onset of abdominal pain, imaging (CT), interventions (corticosteroids, laparotomy), vitals, and laboratory findings.
- Provide higher‑resolution CT figures with arrows.
- Table 1 presents excessive raw data over 13 days without sufficient clinical context. Several parameters lack units and reference ranges, making interpretation difficult. Additionally, some values, such as “95% Oâ‚‚ saturation” despite the presence of a pneumothorax, appear implausible and warrant verification. The authors should summarise key clinical and laboratory trends graphically, such as CRP and leukocyte counts over time.
- Discuss why placental abruption or uterine rupture were ruled out despite contractions and maternal acidosis.
- The manuscript reports a "favourable outcome" but provides incomplete details on the neonatal course. The authors should include comprehensive neonatal outcome data, such as birth weight, Apgar scores at 1 and 5 minutes, umbilical artery pH, the need for neonatal ICU admission, and follow-up status at discharge
- The Discussion section unnecessarily repeats content from the Introduction, particularly the overview of anatomical changes in pregnancy (lines 31–51), which displaces key insights such as the importance of serial examinations and the limitations of ultrasound to the end of the section. The authors should merge repetitive paragraphs to streamline the text and reframe the Discussion to include a critical appraisal of the case. Additionally, they should explicitly compare their findings with the two previously reported cases of small-bowel injuries in pregnancy (e.g., Tasneem 2021, Coleman 2017) to contextualize the diagnostic challenges and outcomes in this clinical scenario.
- The authors’ statement that “no similar cases” have been reported is inaccurate. At least two documented cases of blunt-trauma-induced small-bowel perforation in pregnancy exist.
- Consider revising the title to include “…in Late Pregnancy” to clearly indicate the gestational context of the case and enhance relevance for readers interested in obstetric and surgical complications.
Author Response
Thank you for all your comments. Following it, we have modified the original version of the manuscript.
1.The manuscript’s narrative regarding the clinical course lacks temporal clarity, alternating between references such as “day +4” and “after 72 hours” without a coherent sequence of key events. The authors should provide a clear timeline figure or table that maps hours post-injury against clinical events, including the onset of abdominal pain, imaging (CT), interventions (corticosteroids, laparotomy), vitals, and laboratory findings.
1.Thank you to the reviewer for the suggestion to clarify the temporal evolution of the case. Following that recommendation, we have unified the time terminology used exhibiting days to avoid confusion between days and hours. In addition, we have included a figure (Figure 3) illustrating the clinical course of the patient from admission to discharge, including the vital parameters, the imaging test results, and the medical actions taken.
2.Provide higher-resolution CT figures with arrows.
2.We have contacted the radiology service of our center to try to improve the quality of the CT figures provided in the text. Unfortunately, given that the figures were obtained after a three-dimensional reconstruction carried out in a post-processing program, we have not been able to improve its quality. If the editorial consultant knows any image enhancement or editing software for figures, we would be willing to use it.
3.Table 1 presents excessive raw data over 13 days without sufficient clinical context. Several parameters lack units and reference ranges, making interpretation difficult. Additionally, some values, such as “95% Oâ‚‚ saturation” despite the presence of a pneumothorax, appear implausible and warrant verification. The authors should summarize key clinical and laboratory trends graphically, such as CRP and leukocyte counts over time.
3.Appreciating your suggestion, we have delated table 1 and selected those analytical parameters of most clinical relevance and have decided to represent them graphically in figure (Figure 1). It shows visually the evolution of the principal parameters of interest (leukocytes, lymphocytes, neutrophils and C-reactive protein).
4.Discuss why placental abruption or uterine rupture were ruled out despite contractions and maternal acidosis.
4. After the onset of uterine contractions, a complete physical examination was performed including cardiotocographic recording which showed a reactive pattern with regular uterine dynamic every ten minutes, perceived as painful. On suspicion of onset of active labor, a vaginal examination was carried out (cervical dilatation: 2 cm, thin cervix). Given the cervical conditions, a vaginal ultrasound was not necessary.
The diagnosis of abruptio placentae or uterine rupture are based on physical examination. It includes an assessment of uterine tone by maternal abdominal palpation and assessment of fetal well-being by cardiotocographic recording and abdominal ultrasound.
Specifically, in case of placental abruption, an increased and sustained uterine tone is observed and fetal well-being may be affected in the form of alterations in cardiotocographic recording.
However, when an uterine rupture occurred, uterine palpation usually reveals marked tenderness, especially at the site of the possible rupture, the normal uterine tone is lost and fetal parts can even be palpated directly through the maternal abdomen. In addition, some patients refer shoulder pain from diaphragmatic irritation and vaginal bleeding. Fetal wellbeing is compromised, with frequent loss of fetal wellbeing through alterations in the expected cardiotocographic pattern.
None of the above characteristics were evident in the physical and obstetric examination of the patient, so these obstetric complications were ruled out with high certainty.
5.The manuscript reports a "favourable outcome" but provides incomplete details on the neonatal course. The authors should include comprehensive neonatal outcome data, such as birth weight, Apgar scores at 1 and 5 minutes, umbilical artery pH, the need for neonatal ICU admission, and follow-up status at discharge.
5.Following your recommendations and those of editorial consultant 1 and 2, we have expanded the information provided on the condition of the newborn at birth and his follow-up status at discharge. We have added these details in “Case Presentation” section.
6.The Discussion section unnecessarily repeats content from the Introduction, particularly the overview of anatomical changes in pregnancy (lines 31–51), which displaces key insights such as the importance of serial examinations and the limitations of ultrasound to the end of the section. The authors should merge repetitive paragraphs to streamline the text and reframe the Discussion to include a critical appraisal of the case. Additionally, they should explicitly compare their findings with the two previously reported cases of small-bowel injuries in pregnancy (e.g., Tasneem 2021, Coleman 2017) to contextualize the diagnostic challenges and outcomes in this clinical scenario.
6.Following your recommendation, we have improved the Discussion section by deleting redundant information, particularly related to epidemiological aspects, which was redundant with respect to the “Introduction” section.
We have thoroughly reviewed the two articles indicated (Tasneem 2021 and Coleman 2017), which both form part of our review on the subject and are included as bibliographical references (number 3 and 23).
The first one, entitled “Blunt abdominal trauma in the third trimester: Eight departments, two patients, one survivor” (Tasneem B, Fox D, Akhter S; 2021) describes the case of a 37-year-old patient in her third trimester of pregnancy who arrives at a level 1 trauma center. She was initially evaluated by Emergency Medicine for maternal trauma, by Orthopedics for several fractures including the pubic ramus and sacral ala fractures, as well as by Neurosurgery for a subarachnoid hemorrhage and a subdural hematoma. Subsequently, the patient suddenly became hypotensive with abdominal tenderness suggesting an internal bleeding which was later confirmed. Retroperitoneal and pelvic hematomas were found to be the source of bleeding during an emergency laparotomy. During the surgical procedure, a 2-cm uterine rupture about 4-5 cm below the fundus midline was identified. However, at no point is there any mention of injury to either the large or small intestine.
I don´t if the editorial consultant noticed any confusion in the reading of this article or if he is referring to a different article than the one mentioned.
The same applies to the article entitled “Surgical management of abdominal trauma: Hollow viscus injury” (Coleman JJ, Zarzaur BL; 2017). It focuses on the diagnostic and management of hollow viscus injuries, particularly in blunt trauma, but in a generic way to the general population. I have not found any specific references or recommendations for the pregnant patient group.
7.The authors’ statement that “no similar cases” have been reported is inaccurate. At least two documented cases of blunt-trauma-induced small-bowel perforation in pregnancy exist.
7.Thank you for your suggestion. Both articles are part of our review on the subject.
As I explained in the previous answer, there are publications on blunt trauma in pregnant women but no including intestinal lesion. There are also publications on blunt trauma induced small-bowel perforation in the general population.
However, after an extensive and exhaustive review of the literature, we have not found any publication focuses on blunt abdominal trauma complicated by intestinal ischemia in pregnant women. This is the novelty and distinctive feature of the case presented.
8.Consider revising the title to include “…in Late Pregnancy” to clearly indicate the gestational context of the case and enhance relevance for readers interested in obstetric and surgical complications.
8.Thank you to the reviewer for the suggestion to describe the gestational context of the case. Following it, we have edited the original title to this one: “Intestinal ischemia secondary to blunt abdominal trauma in late pregnancy: A case report of a rare complication with serious implications”.
Round 2
Reviewer 3 Report
Comments and Suggestions for Authors
The authors have adequately addressed all of my comments.